# The Gut–Brain Axis Based on α-Synuclein Propagation—Clinical, Neuropathological, and Experimental Evidence

**DOI:** 10.3390/ijms26093994

**Published:** 2025-04-23

**Authors:** Ryosuke Takahashi, Hodaka Yamakado, Norihito Uemura, Tomoyuki Taguchi, Jun Ueda

**Affiliations:** 1Kyoto University Office of Research Acceleration, Kyoto 606-8501, Japan; 2Department of Therapeutics for Multiple System Atrophy, Kyoto University Graduate School of Medicine, Kyoto 606-8507, Japan; yamakado@kuhp.kyoto-u.ac.jp; 3Department of Neurological Disease Control, Osaka Metropolitan University Graduate School of Medicine, Osaka 545-8585, Japan; nuemura@kuhp.kyoto-u.ac.jp; 4Department of Neurology, Kyoto University Graduate School of Medicine, Kyoto 606-8507, Japan; riverotter@kuhp.kyoto-u.ac.jp (T.T.); junueda1987@kuhp.kyoto-u.ac.jp (J.U.)

**Keywords:** Parkinson’s disease, alpha-synucelin, Lewy body, propagation, the dorsal motor nucleus of the vagus, the olfactory bulb, the enteric nervous system

## Abstract

The cytopathological hallmark of Parkinson’s disease (PD) is a neuronal cytoplasmic inclusion called Lewy body (LB). Lewy bodies are composed of alpha-synuclein (aSyn), a 140 aa protein that is predominantly expressed in the presynaptic terminal and which is implicated in neurotransmitter release. Recently, aSyn was found to propagate from neuron to neuron in a trans-synaptic manner. Although the precise molecular mechanisms are unclear, the propagation of aSyn is believed to play a major role in the progression of Lewy pathology in PD. Neuropathologically, the initial Lewy pathology has been shown to be formed in the dorsal motor nucleus of the vagus (DMV) or olfactory bulb by neuropathological studies. Since the DMV innervates the enteric nervous system (ENS) and LBs are formed in the gut nerve plexuses, it is conceivable that LBs propagate from the gut to the DMV and then to other regions of the brain. In this article, clinical, neuropathological, and experimental evidence supporting or negating the idea that aSyn propagation from the ENS to the brain leads to PD is reviewed. Moreover, the propagation of aSyn seeds through systemic circulation or multifocal generation of aSyn seeds is discussed as a potential alternative scenario for aSyn spreading

## 1. Introduction

The incidence of Parkinson’s disease (PD) is increasing worldwide due to global aging and environmental changes. The number of patients worldwide was 3 million in 1990 and doubled to 6.9 million in 2015. A quarter century later, that is, in 2040, the number is projected to be 14 million. This tendency of patient number increase is called “Parkinson Pandemic”, and immediate action programs for the prevention of PD are needed [1].

PD is a neurodegenerative disorder characterized by hypokinetic motor disturbances, including slowness of movement, resting tremor, and muscle rigidity, mostly caused by dopaminergic neuronal loss in the substantia nigra (SN). Moreover, recent studies have revealed that non-motor symptoms, such as autonomic dysfunction, sleep disorder, hyposmia, and cognitive–psychiatric symptoms, which are caused by neurodegeneration of the non-dopaminergic system, also accompany PD [2].

The pathological hallmark of PD is a cytoplasmic inclusion called Lewy body (LB) [2]. LBs, and their precursor structures, amorphous pale bodies, are observed in neurons vulnerable to PD, including the substantia nigra, the dorsal motor nucleus of the vagus (DMV) and cerebral cortex, suggesting that LBs are closely associated with PD pathogenesis [2] (Figure 1).

In 1997, LBs were shown to be immunopositive and later composed of alpha-synuclein (aSyn) [3]; aSyn is a 140 amino-acid small protein that is abundant in the presynaptic area of neurons. Although its involvement in neurotransmitter release has been suggested, the physiological function of aSyn is yet to be shown [4]. Missense mutations as well as gene duplication/triplication of *aSyn* were shown to be responsible for the highly penetrant autosomal dominant form of familial PD, whose clinical and pathological features are very similar to those of sporadic PD, PD with dementia, and dementia with LB. Although wild-type aSyn is a soluble and naturally unfolded neuronal protein, in vitro protein chemical studies have shown that misfolded aSyn taking beta structure initially forms soluble aggregates including oligomers and protofibrils, eventually becoming insoluble mature fibrils that conform to LB [4]. These lines of evidence strongly suggest that aSyn causes PD [2].

Therefore, aSyn represents a promising target molecule for disease-modifying therapy (DMT) of PD. However, it is well known that more than 50% of dopaminergic neurons in the substantia nigra are lost at the onset of motor symptoms of PD. Even if an effective DMT to eliminate/reduce pathological aSyn species becomes available, it will not be effective when it starts as early as the onset of PD. To achieve success in DMTs for PD, it is essential to diagnose patients at prodromal or earlier preclinical stages. The prodromal stage of PD is characterized by non-motor symptoms including hyposmia, constipation, isolated REM sleep behavior disorder (iRBD), and depression. The lesions responsible for hyposmia are thought to be the olfactory bulb and its circuit, whereas those responsible for constipation, iRBD, and depression are the lower brainstem nuclei [5,6].

Recently, a provocative hypothesis that aSyn accumulates in the enteric nervous system (ENS) caudo-rostrally propagates to the brainstem and then to the substantia nigra in the prodromal stage of PD, reasonably explaining the time course of the above-mentioned prodromal symptoms/lesions, has attracted attention. In other words, “the gut–brain axis” based on aSyn propagation was hypothesized. In this short review, evidence supporting or negating this hypothesis is examined and discussed.

### 1.1. Human Neuropathological and Clinical Evidence for the Enteric Nervous System as the Origin for PD

In 1984, Qaulman et al. first reported LB formation in the enteric nervous system (ENS). They found LB pathology in the myenteric nerve plexus of the esophagus in two patients with esophageal achalasia and two patients with PD, and suggested their correlation with gastrointestinal movement disturbance [7]. Moreover, a neuropathological analysis of seven patients with PD revealed LB formation in both the submucosal and myenteric plexuses of the digestive tract, most prominently in the lower esophagus, of all the cases examined [8].

A recent study examined LB formation in 518 autopsy-confirmed cohorts of older (>65 years old) Japanese people from the Brain Bank for Aging Research (BBAR) established in a geriatric hospital. Results show that one-third of the cases examined (178 out of 518 cases, 34%) exhibited Lewy pathology, of which 78 (43.8%) exhibited pathology in the esophagus [9]. On the other hand, Heiko Braak proposed an innovative hypothesis about the origin and progression pattern of LB pathology in the PD brain, which is well-known as “Braak’s hypothesis” [10]. According to Braak’s hypothesis, initial LB lesions are formed in the DMV and olfactory bulb and consecutively propagate to the upper brain structure. It is also emphasized that only projection neurons with disproportionately long, thin, and weakly myelinated or non-myelinated axons tend to form the Lewy pathology. In contrast, interneurons and projection cells with short axons or those with heavily myelinated axons do not form LB lesions [10]. Motor symptoms emerge when Lewy pathology reaches the substantia nigra. Based on these observations, a two-hit hypothesis was proposed. According to this hypothesis, neurotropic pathogens such as viruses or toxins are postulated to interact with the intestinal and olfactory epithelia, leading to the formation of initial LB lesions in the GI tract and olfactory bulb [11]. Braak further claimed that pathological processes generally progress caudorostrally through the nerve nuclei of the lower brainstem, including the DMV, lower raphe nuclei, magnocellular nuclei of the reticular formation, and locus coeruleus into midbrain tegmental nuclei, including the pedunculopontine nucleus and dopaminergic projection neurons of the substantia nigra and nuclei of the upper raphe system, and non-cortical centers of the forebrain, including the amygdala, hypothalamic tuberomammillary nucleus, magnocellular nuclei of the basal forebrain, and midline and intralaminar nuclei of the thalamus, and then reach the cerebral cortex starting from the transentorhinal and entorhinal regions, Ammon’s horn, cortical visceromotor areas, and, finally, the entire neocortex. This claim significantly contributed to the formation of the “gut-first” hypothesis [5]. To further develop this concept, body-first and brain-first PD subtypes were proposed by Horsager and Borghammer et al. using in vivo multimodal imaging [12]. In body- and brain-first PD, the initial aSyn aggregates are hypothesized to be formed in the enteric or peripheral autonomic nervous system and the brain and propagate from the peripheral autonomic nervous system to the brain and from the brain to the peripheral autonomic nervous system, respectively. Importantly, they hypothesized that iRBD is a prodromal phenotype of the body-first subtype. They quantified the neurological function of patients with de novo PD using functional neuroimaging: 24 RBD-negative (PDRBD–) and 13 RBD-positive (PDRBD+) cases and a comparator group of 22 iRBD patients. They used 11C-donepezil PET/CT to assess parasympathetic innervation, 123I-metaiodobenzylguanidine (MIBG) scintigraphy to measure cardiac sympathetic innervation, neuromelanin-sensitive MRI to measure the structure of locus coeruleus pigmented neurons as a surrogate marker for iRBD-responsible regions, and 18F-dihydroxyphenylalanine (FDOPA) PET to assess striatal dopamine storage capacity. Colon volume and transit time were assessed using CT tomography and radiopaque markers. Compared to PDRBD− patients, PDRBD+ and iRBD patients show reduced mean MIBG uptake and colon 11C-donepezil standard uptake values. The PDRBD+ group shows a tendency towards a reduced mean MRI locus coeruleus: pons ratio in comparison to the PDRBD− group. Relative to the other groups, the PDRBD+ group had enlarged colon volumes and delayed colonic transit times. The combined iRBD and PDRBD+ patient data were compatible with the body-first subtype, characterized by the early reduction in cardiac MIBG and 11C-colonic donepezil signals, followed by a decline in striatal uptake of FDOPA. In contrast, the PDRBD− data were compatible with the brain-first subtype, characterized by the primary loss of striatal FDOPA uptake, followed by a secondary impairment of peripheral autonomic nervous function.

This hypothesized aSyn progression pattern was supported by the gradients of aSyn pathology in the whole body, represented by caudo-rostral and amygdala-centered patterns based on Finnish brain bank data [13]. Moreover, Borghammer et al. recently proposed a synuclein origin and connectome (SOC) disease model of Lewy body disorders [14]. This model is based on the hypothesis that in the majority of patients, the first Lewy pathology arises at a single location and propagates from the initial lesions. The most common sites of origin are the enteric nervous system (ENS) and the olfactory bulb/amygdala. The SOC model predicts that Lewy pathologies in the ENS result in a clinical body-first subtype characterized by autonomic symptoms and iRBD at the prodromal stage. In contrast, the Lewy pathology of the olfactory system results in a brain-first subtype without autonomic dysfunction or iRBD before diagnosis. According to the SOC model, patients with the body-first subtype are older, more likely to develop symmetric dopaminergic degeneration, and are at an increased risk of dementia than patients with the brain-first subtype. However, the SOC model has been challenged by a recent imaging study that focused on the asymmetry of dopaminergic degeneration [15]. Moreover, strong pathological evidence that cardiac sympathetic denervation starts from postganglionic neurons contradicts the SOC model [16]. In the SOC model, the cardiac sympathetic nerves are interpreted to be an intermediate site of aSyn propagation from the gut to the brain: aSyn aggregates are hypothesized to be transported from the intermediolateral nuclei in the spinal cord to the sympathetic ganglia, then to the postganglionic sympathetic nerve. However, an autopsy study of the cardiac sympathetic nerves in PD shows that phosphorylated aSyn accumulates in the distal parts of the cardiac sympathetic axons before the cell bodies in the sympathetic ganglia [17]. Moreover, it was reported that LB pathology in the sympathetic ganglia preceded that in the nucleus of the intermediolateral column of the thoracic cord [18]. Furthermore, in an autopsy case, aSyn pathology was restricted to the heart and stellate ganglia [19]. Further studies are required to validate this model.

### 1.2. Gut Microbiota and aSyn

The involvement of gut microbiota has attracted attention as a cause of aSyn accumulation in the enteric nerve plexus. Bacteria may invade submucosal tissue due to hyperpermeability of the gut epithelia, that is, “leaky gut”. Alternatively, dysbiotic gut microbiota can induce inflammation by stimulating Toll-like receptors expressed on the membranes of epithelial cells or immune cells. Such inflammatory reactions may promote aSyn upregulation and misfolding in enteric nerve plexuses, leading to the formation of aSyn aggregates. It has been hypothesized that these aggregates also induce inflammation, forming a vicious circle [20]. Akkermansia, which increases in the gut microbiota of PD patients, degrades the mucous layer and causes hyperpermeability of the gut epithelia, inducing “leaky gut” [21]. On the other hand, short chain fatty acid (SFCA)-producing bacteria, including Faecalibacterium, Roseburia, and Agathobacter, are reported to be decreased in the gut of PD patients. These changes may decrease the number of regulatory T cells activated by SCFA, possibly inducing inflammation [21]. The above-mentioned hypothesis that changes in the gut microbiota of patients with PD may lead to inflammation of the gastric wall and promote the expression and aggregation of aSyn should be examined in further studies.

It has also been noted that enteroendocrine cells (EECs) with characteristics of sensory neurons located in the gut epithelia are innervated by the sensory branch of the vagus nerve and express aSyn [22]. It is also interesting to determine whether aSyn can propagate from the EEC to the brain. However, since the sensory branch of the vagus nerve innervates the EEC and projects to the solitary nucleus, this route will not contribute to LB formation in the DMV.

### 1.3. Epidemiological Evidence for the Enteric Nervous System as the Origin for PD

Evidence supporting the gut–brain axis has been obtained from epidemiological studies. A study examined the risk of PD in patients with peptic ulcers who underwent truncal vagotomy and super-selective vagotomy as well as in non-treated subjects based on a Danish cohort composed of people registered in the Danish Civil Registration System; the number of patients with truncal vagotomy, super-selective vagotomy, and control subjects were 5300, 5900, and over 60,000, respectively. Twenty years after vagotomy, the risk of PD was decreased in patients who underwent truncal surgery compared with the general population cohort. The risk of PD in patients who underwent super-selective vagotomy was similar to that in the general population.

These results suggest that the vagal nerve plays a critical role in the pathogenesis of PD and are consistent with the idea that aSyn aggregates in the ENS can be transported to the brain via the vagus nerve [23]. A subsequent Swedish cohort study also suggested that truncal vagotomy may reduce the risk of PD in later years [24].

### 1.4. The Experimental Evidence for the Enteric Nervous System as the Origin for PD

Does experimental pathology provide evidence to support the idea that aSyn may reach the brain from the ENS via the vagus nerve?

Although Braak et al. originally postulated that aSyn can propagate from their initial lesions, including the DMV and olfactory bulb, they did not prove aSyn propagation experimentally [10]. However, strong evidence for aSyn propagation has been obtained from studies on human brain autopsy samples. Fetal ventral midbrain tissuetransplantation was initiated in Sweden in 1989 [25]. In the brains of patients with PD who died 10 years after the transplantation of fetal ventral midbrain tissues, aSyn-positive LB-like inclusions were found in the fetal dopamine cells of the graft [26,27]. This strongly suggests that aSyn aggregates formed in the brains of patients propagated to the transplanted fetal dopamine neurons. To test this idea, Luk and Lee et al. produced recombinant aSyn, incubated it into fibrils (preformed fibril: PFF), and inoculated sonicated and fragmented fibrils into the wild-type mouse brain. Inoculation leads to cell-to-cell transmission of pathological aSyn in interconnected neuronal nuclei [28]. Masuda-Suzukake and Hasegawa et al. injected fibrils made from human aSyn into wild-type mouse brain, which resulted in Lewy body/neurite-like pathology in various brain regions connected by neural networks. Interestingly, human aSyn disappeared in about a week after injection, and the aSyn aggregates formed in the mouse brain were shown to be composed of endogenous mouse aSyn. These results indicate that aSyn fibrils have prion-like properties, and pathological misfolded aSyn serves as the template for wild-type aSyn to convert to pathological fibrillar aSyn [29] (Figure 2).

Although the mechanisms underlying aSyn fibril release and uptake are largely unknown, we propose that aSyn fibrils are taken up via macropinocytosis, which is regulated by neuronal electrical activity. Perampanel, an AMPA receptor antagonist, inhibits the electrical activity and uptake of aSyn PFFs in cultured mouse cortical neurons, blocking aSyn propagation in vivo [30].

aSyn fibrils were inoculated into the stomach with the aim of proving aSyn propagation from the gut to the brain. The initial report on the inoculation of different forms of aSyn, that is, monomer, oligomer, and fibrillar species, into the rat intestine showed that they were transported via the vagus nerve to the DMV in a time-dependent manner [31]. However, there is no description of aSyn propagation beyond the DMV. When we injected fragments of aSyn PFF into the pyloric region of the stomach, aSyn aggregates formed in the DMV 30–45 days after the injection. Inoculation of aSyn monomers did not cause aggregate formation in the DMV. Aggregate formation was completely abolished when the vagus nerve was resected before aSyn fibril inoculation, indicating that aSyn fibrils traveled from the stomach to the vagus nuclei via the vagus nerve. Consistent with this hypothesis, aSyn fibrils were observed in the submucosal and myenteric plexuses of the stomach. However, aSyn aggregates were not formed in the upper brainstem structures beyond the DMV 12 months after inoculation [32]. Another report showed that inoculation of aSyn fibrils into the intestinal wall led to the formation of aSyn aggregates in the DMV and locus coeruleus (LC) [33]. In contrast, it has been reported that inoculation of aSyn fibrils into the rat stomach wall results in aggregate formation in the DMV, LC, substantia nigra pars reticulata, amygdala, piriform cortex, entorhinal cortex, retrosplenial cortex, granular cortex, and olfactory bulb in the central nervous system. aSyn aggregate pathologies were also observed in peripheral nerves and tissues, including the celiac ganglia and cervical ganglia of the sympathetic trunk, intermediolateral nucleus of the spinal cord, myocardial ganglia, and sympathetic nerves in the heart, skin, and muscle. These extensive pathologies are hypothesized to be formed by aSyn propagation via both the sympathetic and parasympathetic nervous systems [34]. These widespread aSyn lesions were observed in adult (10–12 months old) and old rats (18 months old), but not in young rats (3 months old). Interestingly, the same group reported aSyn propagation from the stomach to the substantia nigari zona reticulata in four-month-old, young bacterial artificial chromosome (BAC)-*SNCA* transgenic rats overexpressing the full-length human *aSyn* gene under the control of endogenous human regulatory elements [35]. This suggests that the expression level of aSyn may increase with age due to the disturbance of proteostasis. However, the injection of aSyn PFF into the stomach wall of BAC-*SNCA* transgenic mice, which were created with the same BAC construct as the BAC-*SNCA* transgenic rats, gave rise to different results. aSyn propagation was mildly enhanced to reach the nucleus ambiguous, but not any upper structures, including the LC and substantia nigra pars reticulata [36]. The impact of aSyn expression on aSyn propagation should be further examined. Another study reported that inoculation of aSyn PFF into the gastric wall leads to extensive aSyn propagation to the substantia nigra pars compacta and olfactory bulb, resulting in locomotor disturbance and hyposmia, respectively [37]. As described above, the level of aSyn propagation varies significantly according to the individual reports. Numerous factors affect the propagation, including the preparation, size, and amount of fibrils, site of injection, age of animals, and animal species. These points should be carefully examined and standardized to correctly interpret the results. Despite these inconsistencies in the results, the propagation of aSyn from the gut to the DMV has been reported in all reports cited in this review, strongly suggesting the gut–brain route via the vagus nerve. In contrast, aSyn propagation from the gut to the substantia nigra pars compacta has not been demonstrated, except for a single report [37].

To examine whether aSyn propagates from the gut to the brain. several points should be made clear. It has been pointed out that direct neural connections between the DMV and LC as well as the LC and substantia nigra pars compacta were not demonstrated [38]. The precise connections from the DMV and LC to the upper structures should be clarified. Moreover, most studies to date have been conducted in rodents. Experiments using non-human primates (NHP), whose neuroanatomy is similar to that of humans, may give rise to different results. Notedly, we have reported that inoculation with aSyn PFF to the olfactory bulb leads to the formation of aSyn aggregates in the DMV, presumably via the amygdala [39].

Taken together, the gut-to-brain propagation hypothesis proposed by Braak et al. as the cause of Parkinson’s is yet to be proven experimentally [10].

### 1.5. The Role of Systemic Circulation in Seed Spreading

Recently, bidirectional gut-to-brain and brain-to-gut aSyn propagation has been reported in non-human primates [40]. In this study, Arotcarena and Bezard et al. examined the pathological consequences of intrastriatal or enteric injections of α-synuclein-containing Lewy body extracts from monkeys with Parkinson’s disease. Patient-derived α-synuclein aggregates can induce nigrostriatal lesions and ENS pathology after duodenal or striatal injections in a non-human primate model. These results suggest that the spread of α-synuclein pathology might be bidirectional, that is, either in a “body-first” or “brain-first” manner. Surprisingly, under their experimental conditions, α-synuclein pathological lesions were not observed in the vagal nerve. This study does not support the hypothesis that α-synuclein pathology is transmitted through the vagus nerve and the DMV. These results suggest that systemic circulation plays a major role in the bidirectional propagation of endogenous α-synuclein between the gut and the brain [40]. Recently, an extremely small amount of aSyn aggregates was detected in the serum of human PD patients using IP-RT-QIUC, an efficient protein amplification system [41,42]. This finding suggests that aSyn propagation may be mediated through systemic circulation in addition to neural connections.

### 1.6. Multifocal Generation of aSyn Seeds

Another important possibility is that aSyn pathology located in the ENS and DMS would be caused by multifocal formation of aSyn aggregates or seeds rather than propagation: aSyn aggregates are alternatively called seeds, since they can induce the misfolding and aggregation of normal aSyn in a prion-like manner (Figure 1). Although aggregate formation from wild-type aSyn is thought to be a rare event, mutant aSyn is more prone to misfolding and aggregate formation.

The clinical features of patients with SCNA duplication mutations include L-dopa-responsive parkinsonian motor symptoms and non-motor signs or symptoms such as depression, hallucinations, RBD, and dysautonomia. Moreover, the neuropathological findings of these cases are characterized by LB formation in the substantia nigra pars compacta and locus coeruleus, suggesting that SCNA duplication mutation cases are almost indistinguishable from those of idiopathic PD, both clinically and neuropathologically [43].

Carriers of *aSyn* A53T mutations have also been reported to exhibit motor and non-motor features similar to idiopathic PD, with evidence of a more severe nigrostriatal denervation [44]. These data suggest that the expression of the *aSyn* A53T mutation under the control of its native promoter may provide an excellent animal model that recapitulates the signs and symptoms of PD.

Based on this idea, we created bacterial artificial chromosome transgenic mice harboring *SNCA* and its gene expression regulatory regions to maintain the native expression pattern of aSyn. Furthermore, to enhance the pathological properties of α-synuclein, we inserted an A53T mutation into *SNCA* and two single-nucleotide polymorphisms identified in a genome-wide association study in Parkinson’s and a Rep1 polymorphism, all of which are causal of familial Parkinson’s disease or increase the risk of sporadic Parkinson’s disease. Remarkably, this model recapitulates prodromal symptoms and signs of PD, including an RBD-like phenotype, hyposmia, decreased intestinal movement, and mild dopamine neuron-specific cell loss. Neuropathological findings reveal truncated, oligomeric, and proteinase K-resistant phosphorylated forms of aSyn in the olfactory bulb, cerebral cortex, striatum, and substantia nigra without fibril formation, suggesting that fibril-dependent propagation is unlikely to occur in this mouse model [45]. Rather, independent multifocal generation of aSyn seeds is a plausible scenario. Moreover, multiple prodromal symptoms such as hyposmia and RBD, whose responsible regions are the olfactory circuit and lower brainstem, manifest at the age of 5–9 months without fibril formation, suggesting multifocal seed generation.

## 2. Discussion

aSyn propagation from the gut to the brain through neural connections is an attractive hypothesis to explain the aSyn spreading pattern in PD, as Braak et al. have originally proposed based on autopsy cases, and Borghammer et al. have further developed based on clinical PD and iRBD cases [10,11,12,13,14]. However, there appears to be lines of counterevidence to negate this hypothesis, and the presence of the gut–brain route is controversial. How should the current situation be reconciled? First, the connection between the ENS and the brain nuclei, especially the ENS to the substantia nigra, should be fully examined. Second, the relationship between the centripetal process of Lewy pathology at a single neuronal level (i.e., distal axons and terminals to the cell soma) and the stereotypic LB pathology spreading pattern should be elucidated. Uchihara claimed that retrograde degeneration in hyperbranching axons due to aSyn aggregate accumulation in distal axons, as observed in nigral dopaminergic neurons and cardiac sympathetic neurons, is a typical neurodegenerative process in Lewy body disease, consistent with multifocal aSyn seed generation in distal axons [38]. However, the cellular LB formation process is also theoretically consistent with retrograde propagation of aSyn seeds. Moreover, how multifocal seed generation leads to stereotypic LB pathology spreading patterns needs to be explained, although a number of autopsy cases showing atypical LB pathology spreading patterns have been reported [38]. Remarkably, most iRB cases with prodromal LB disease exhibit cardiac sympathetic nerve denervation as assessed by MIBG scintigraphy, which is difficult to explain by the random multifocal origin of LB disease. The propagation of aSyn seeds through systemic circulation is another possibility; however, such a propagation pattern, as well as multifocal seed generation, does not appear to be consistent with the caudo-rostral LB spreading pattern shown by Braak.

## 3. Conclusions and Future Directions

Taken together, current evidence suggests that aSyn propagation through neural connections and/or systemic circulation or multifocal generation of aSyn aggregates is a potential underlying pathophysiology of the gut–brain axis in PD (Figure 3). How these mechanisms or unknown mechanisms of aSyn propagation contribute to the caudo-rostral progression of LB pathology in PD should be further explored.

## Figures and Tables

**Figure 1 ijms-26-03994-f001:**
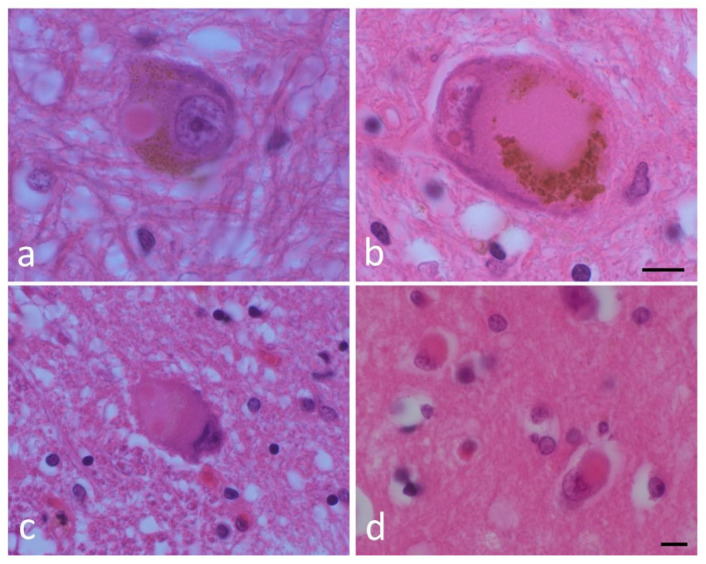
Lewy bodies (LBs) and a pale body. (**a**) An LB and (**b**) a pale body in a dopamine neuron in the substantia nigra. (**c**) An LB in the dorsal motor nucleus of the vagus. (**d**) A cortical LB. Hematoxylin and eosin (H&E) staining of brain tissues. Scale bar: 5 μm. (By courtesy of Dr. Toshiki Uchihara).

**Figure 2 ijms-26-03994-f002:**
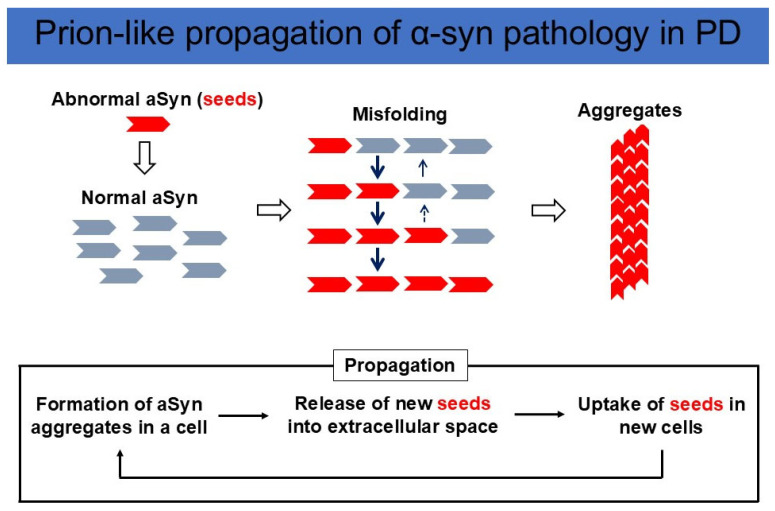
Prion-like propagation of aSyn pathology in PD. (Upper panel) Misfolded aSyn works as a seed, convert wild-type aSyn into an abnormal conformation, and amplify aggregates. (Lower panel) New seeds are released into the extracellular space and taken up by neighboring neurons, followed by further α-syn misfolding and aggregation. This cycle triggers the propagation of an abnormal aSyn pathology.

**Figure 3 ijms-26-03994-f003:**
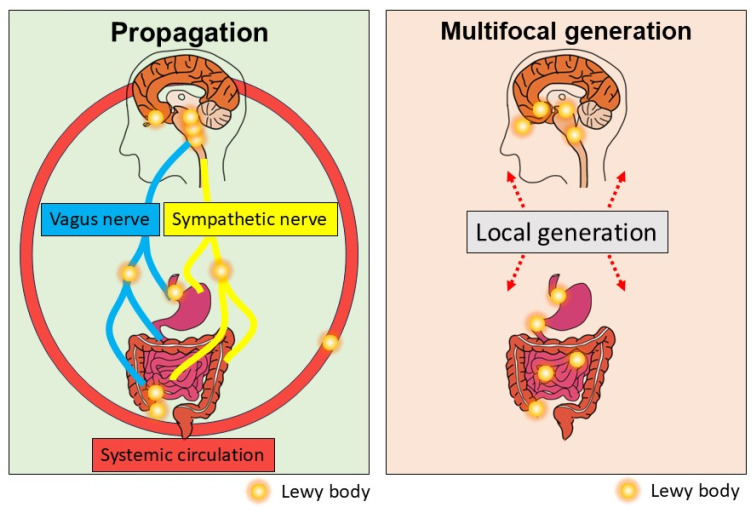
Current understanding of the gut–brain axis in PD. aSyn aggregates formed in the brain or gut could bidirectionally propagate through the vagus nerve or the sympathetic nerve or systemic circulation.

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
