# Peer review of "The Gut–Brain Axis Based on α-Synuclein Propagation—Clinical, Neuropathological, and Experimental Evidence"

_ijms, 2025, doi:10.3390/ijms26093994_

Round 1
Reviewer 1 Report
Comments and Suggestions for Authors
This manuscript offers a comprehensive and timely review of the gut-brain axis in Parkinson’s disease (PD), particularly focusing on the propagation of alpha-synuclein (aSyn) from the enteric nervous system (ENS) to the central nervous system. The authors provide a well-balanced discussion that integrates clinical, neuropathological, experimental, and epidemiological evidence. The inclusion of both supporting and contradictory findings adds credibility and objectivity to the narrative. Totally well-written.
One minor suggestion for key Figure 2 is to consider using color-coded arrows or labels to distinguish the proposed pathways (vagus nerve, sympathetic nerve, systemic circulation), which would further aid clarity.
Author Response
Comment 1: This manuscript offers a comprehensive and timely review of the gut-brain axis in Parkinson’s disease (PD), particularly focusing on the propagation of alpha-synuclein (aSyn) from the enteric nervous system (ENS) to the central nervous system. The authors provide a well-balanced discussion that integrates clinical, neuropathological, experimental, and epidemiological evidence. The inclusion of both supporting and contradictory findings adds credibility and objectivity to the narrative. Totally well-written.
Response 1: Thank you very much. We really appreciate the reviewer's very positive comments.
Comment 2: One minor suggestion for key Figure 2 is to consider using color-coded arrows or labels to distinguish the proposed pathways (vagus nerve, sympathetic nerve, systemic circulation), which would further aid clarity.
Response 2: We appreciate the reviewer's kind and insightful comment. We revised Figure 2(Figure 3 in the revised manuscript) according to the reviewer's advice.
Reviewer 2 Report
Comments and Suggestions for Authors
The authors overviewed the significance of the gut-brain axis on a-Synuclein (aSyn) propagation in Parkinson’s disease (PD) based on clinical, neuropathological and experimental evidence. They summarized using Figure 2 that a-Synuclein propagation through neural connection and/or systemic circulation or multifocal generation of aSyn aggregates is a potential underlying pathophysiology of the gut-brain axis in PD. The manuscript reviewed in detail molecular mechanisms in aSyn propagation associated with PD is very useful for neuroscientists as well as PD researchers. This review paper is appropriate for publication in International Journal of Molecular Sciences. The reviewer has some comments.
- (line 55) Is “aSyn Therefore, aSyn represents a promising target molecular for disease-modifying therapy of PD” correct? Is the first word “aSyn” needed? Should (DMT) be inserted after a phrase “disease-modifying therapy”?
- The authors could show neuropathological findings of aSyn deposition in the central nervous system, such as substantia nigra or the dorsal motor nucleus of the vagus, and the enteric nervous system. These neuropathological images must be impressive.
Author Response
Comments 1: The authors overviewed the significance of the gut-brain axis on a-Synuclein (aSyn) propagation in Parkinson’s disease (PD) based on clinical, neuropathological and experimental evidence. They summarized using Figure 2 that a-Synuclein propagation through neural connection and/or systemic circulation or multifocal generation of aSyn aggregates is a potential underlying pathophysiology of the gut-brain axis in PD. The manuscript reviewed in detail molecular mechanisms in aSyn propagation associated with PD is very useful for neuroscientists as well as PD researchers. This review paper is appropriate for publication in International Journal of Molecular Sciences.
Response 1: We really appreciate the reviewer's positive comments.
Comments 2: (line 55) Is “aSyn Therefore, aSyn represents a promising target molecular for disease-modifying therapy of PD” correct? Is the first word “aSyn” needed? Should (DMT) be inserted after a phrase “disease-modifying therapy”?
Response 2: Thank you for usuful comments. We changed the senstence as follows: (line 63 in the revised manuscript)“Therefore, aSyn represents a promising target molecular for disease-modifying therapy (DMT) of PD”
Comments 3: The authors could show neuropathological findings of aSyn deposition in the central nervous system, such as substantia nigra or the dorsal motor nucleus of the vagus, and the enteric nervous system. These neuropathological images must be impressive.
Resonse 3: We apprecaite the rviewer's insightful comments. We added a new Figure showing photos of Lewy bodies and a pale body in the substantia nigra, the dorsal motor nucleus of the vagus and the cerebral cortex as Figure 1 in the revised manusript.